# Single Mothers' Perspectives on the Combination of Motherhood and Work

**Dries Van Gasse *** and **Dimitri Mortelmans**

Department of Sociology, University of Antwerp, Sint-Jacobsmarkt 2, 2000 Antwerp, Belgium;
Dimitri.Mortelmans@uantwerpen.be
* Correspondence: Dries.vangasse@uantwerpen.be

**Abstract:** This study aims to define the perspectives taken by single mothers when combining work and motherhood in a stressful work–life constellation. One of the challenges for single mothers after divorce is to find a work–life balance in their single-parent family system. Regarding work-life balance, we take a General Strain Perspective, describing the work-life conflict as a combination of financial strain and role strain. We argue that both strains are the most problematic for single mothers in comparison to their married and/or male counterparts, as both finances and parenthood ideologies are more under pressure. For this reason, we explore how single mothers coped with this strain, answering the question: 'Which perspectives on the combination motherhood and work do single mothers take in their attempt to balance role strain and financial strain after divorce?' To answer this research question, we used a qualitative approach, based on 202 in-depth interviews with single mothers in Belgium. These interviews involved two groups: A primary research population of 13 single mothers and an elaborative research population of 189 single mothers. Timelines were used to structure the single mothers' narratives. The analysis resulted in the contruction of a typology of four different perspectives based on how single mothers dealt with maternal role strain and financial strain: the re-invented motherhood perspective, the work-family symbiosis perspective, the work-centered motherhood perspective and the work-family conflicted perspective. We found that perspective of single mothers in their work-life strain can be described by the flexibility and/or strictness in either their motherhood ideology and/or their work context. These results point at the needs for policymakers, employers, and practitioners to focus on initiatives improving the work–life balance of single mothers by reducing financial and role strains.

**Keywords:** work–life balance; motherhood ideology; role strain; financial strain; separation; divorce; single motherhood; general strain theory; social construction of motherhood; workplace flexibility; single mothers

## 1. Introduction

The combination of work and family life is a challenge for many people in society today (Kalliath and Brough 2008). However, this struggle is not the same for everyone. In comparison to their married, cohabiting, or childless counterparts, single parents have the worst work–life balance (Van den Eynde et al. 2019). This is partially due to the need for an increase in time spent at work after divorce or separation[1] (Thielemans and Mortelmans 2019). Moreover, this need is especially apparent

---

[1] The study present narrates about the work–life conflicts of single mothers. Throughout the article, we use divorce and separation to refer to the end of a cohabitation and/or marriage with a romantic partner of any sexuality. More details about the research population can be found in the method section.

for women. The explanations for this differ, with some arguing that it is based on anticipation, others understanding it in terms of precedent, and yet others as selective antecedent.

First, Thielemans and Mortelmans (2019) coined the term 'anticipation' to explain this increase in time at work before actual separation. They argue that women in particular attempt to reduce their financial dependence before divorce. This can be linked to the issue of financial need and is tied to the financial strain perspective. Second, we can see this dynamic as a precedent before divorce as authors have argued that labor market participation causes stress in the roles played by parents in the family system. This perspective of the increase of female labor force participation as a precedent of divorce is tied to the gendered institution perspective (Killewald 2016). The stress related to the introduction of new roles in the household can cause conflict when women want to renegotiate task specialization after entering the labor market. Therefore, an increase of female labor force participation may *precede* divorce as it moderates marital distress. Third, as men are most often the dominant earner, we can also frame the increase of women's working time as a selective antecedent. Due to the higher work intensity of men in general, there is less room to increase work time from a male perspective, whereas an increase in work time in women is more visible in any move towards independence as their time on the labor market is generally lower than for their male counterparts (Ciccia and Bleijenbergh 2014). This general increase in work time and the lack of a partner with whom to share household tasks has been found to generate a general increase in work–life conflict from the viewpoint of work (Bakker and Karsten 2013). Therefore, the increase that is observed in female labor force participation can be framed as a *selective antecedent* as, contrary to men, women are less likely to be already full-time at work in the dual-partner household.

Nevertheless, work–life conflict is not only a problem with respect to work. Therefore, research should not only focus on the financial strain aspect of the problem (Nomaguchi and Milkie 2017), in which the economic and work situation is the focal point of the analysis (Hughes and Parkes 2007). This research focus assumes that the main reason behind the work–family conflict of single parents is the pressing financial needs required to support a family after and while recovering from divorce or separation. However, with respect to single parents, the issues are broader than sheer economic security, as they must also transition to a new role as a single parent after divorce (Van Gasse and Mortelmans 2020a). Therefore, it is also of importance to examine the work–family balance from a role strain perspective (Kelly and Voydanoff 1985).

From this older theoretical perspective, work–family balance is regarded as entailing an ideal combination of an individual's role as an employee and their social role as a parent (Moen and Dempster-McClain 1987). More recently, this framework has been the basis of studies that relate gender ideologies to coping strategies at the workplace. Somech and Drach-Zahavy (2007), for example, argued that there are eight coping strategies used by parents in the workplace (each of which is either more work focused or more family focused), and the choice of strategy is mainly divided along gender lines. Parental role strain is therefore best to be described from the perspective of General Strain Theory. In this perspective, the strain on parents derives from the inability to achieve positively valued goals (e.g., the parenthood ideals that are defined by parents) (Agnew et al. 2002). Strain increases due to the more complex family system single parents live in and it affects the experiences and behavior of people. As women's parenthood ideals are stricter than men's, Bagger et al. (2008) argue that work–life balance is a gendered issue, as the role conflict is often worse for mothers than for fathers.

Nevertheless, research increasingly indicates that fathers also suffer from work–life issues as they are reframing fatherhood ideals (Daly et al. 2008). We can, however, argue that the discourse for fathers' work–life balance is essentially different due to the historical context in which the work–life conflict arose. Whereas women traditionally took up the primary role as caretaker, men are traditionally seen as breadwinners (Duckworth and Buzzanell 2009). Therefore, the shift women generally make is a shift from caretaker to a shared role as caretaker/breadwinner and men shifting from breadwinner to breadwinner/caretaker. As a result, the challenges for men and women are different due to the

core difference in meaning of the shift that is made in both genders after divorce or separation (e.g., Rudman and Mescher (2013) argue that men are more penalized than women when they ask for flexibility at work due to family care duties.). We can thus argue that both men and women have difficulties with the combination of role strain and financial strain, but as the discourse inherently differs, a primary focus should be chosen.

For this reason, this study focuses on how single mothers cope with role strain while being active in the labor market. As we argued, the role strain of single mothers is a result of the inability to achieve self-defined goals. Therefore, we use the concept of motherhood ideologies: The constructed frameworks of norms, values, and beliefs of single parents concerning parenthood (Elliott et al. 2013). The aim is to combine the complex issue of role strain with the already known issue of financial strain in single parents. Our research question is: 'Which perspectives on the combination motherhood and work do single mothers take in their attempt to balance role strain and financial strain after divorce?'

We have chosen to pay specific attention to single *mothers* as, from a financial strain perspective, and as authors such as Thielemans and Mortelmans (2019) have argued, the increase in time at work, and thus a shift in roles, is the most visible after divorce. However, the traditional labor market appears to be family unfriendly at its core and all single parents have to navigate through the demands of this family unfriendly world (Strachan and Burgess 1998). Working hours do not match school hours and single mothers do not have a partner to share tasks with (Van den Eynde et al. 2019). The struggle faced by single fathers may be similar, but less urgent by comparison because of the financial differences. Therefore, we will leave this focus group to future research that should focus to the gender-specific issues of single fathers' work life balance as well. We do explicitly look into single motherhood after divorce, although other consequences of single parenthood, such as single parenthood by choice or due to bereavement also occur in Belgium (Van Gasse and Mortelmans 2020b; Walsh and McGoldrick 2013). The reasoning behind studying single motherhood after divorce or separation is due to the well-documented increase of stress levels in the period wherein single mothers reorganize their lives to start a single-parent family, illustrating the role and financial strain in this transition process (Booth and Amato 1991; Van Gasse and Mortelmans 2020a).

## 2. Literature Review

### 2.1. Parenthood Ideologies

Parenthood ideologies are social constructed, normative frameworks, with which parents form their general expectations and aspirations about parenting (Taylor 2011). This construction of parenthood ideologies occurs on an individual and collective level. This was illustrated by Purewal and van Den Akker (2007), who found that some elements of parenthood ideologies are universal, while others are defined by gender, age, cultural background, and parity. Baluta (2014) found that the universal aspects of gendered parenthood ideologies can even be institutionalized in policies and thus have an impact on how work and life in general are organized. Examples of such institutionalized parenthood ideologies are apparent in parental leave systems in countries that target mothers assuming they will want a more nurturing role in the family system than would fathers, who often remain the traditional 'male breadwinners' (Haas 2003).

This institutionalization is closely related to the theory of 'doing gender' that explains how these socially constructed frameworks may be internalized (Bruni et al. 2004). Doing gender implies that we act according to the roles that are prescribed through the socially constructed expectations we have about gender (West and Zimmerman 1987). Although certain policies may seem only to prescribe role behavior at a higher level, these norms and values about parenthood are often internalized. For example, reviewing decades of research on women's careers, Phillips and Imhoff (1997) found that women are historically nudged towards a more family-centered attitude in comparison to men and, therefore, gender differences in the labor market can partially be explained from this doing gender perspective. On the other side of the gender axis, Glauber and Gozjolko (2011) demonstrated how the

traditional fathering role entails more activity on the labor market and how more egalitarian fathers stumble on these socially established expectations.

Although it is often argued in the literature on gender inequality in the labor market that this is the result of socially constructed parenthood ideologies, this is not the only reason. For example, there are also different expectations in terms of childcare. In general, mothers adhere more strictly to the ideology of playing the nurturing role in parenthood (Allen and Hawkins 1999). This is partially reinforced at an institutional level (e.g., through the parental leave system) but may also be reinforced through individual mechanisms, such as 'maternal gatekeeping', which sometimes excludes the other parent from performing household or parenting tasks (Puhlman and Pasley 2017). Maternal gatekeeping means that women often take on more parental responsibility, as they feel it is part of their work–family trade-off in the household (Radcliffe and Cassell 2015). This not only means that even women in dual-earner families usually take on the care tasks, but that many are also not that eager to give up this role. In this way, their own parental ideology of nurturing complements fathering ideologies that are more focused on the financial support of families.

If we consider this theory on parenthood ideologies in relation to single parenthood, single mothers find themselves in a very difficult position after divorce. Not only do they have to grapple with the strain on their motherhood role, they also have to take on the role of breadwinner, being the sole earner in the family. As a result, they must often increase their work time to gain financial independence from their former partner (Thielemans and Mortelmans 2019). Thus, they not only face the practical problem of finding a new balance between their roles as employee and parent, but also deal with their own ideological constraints. As such, single mothers need to think outside the box when re-negotiating their work and family responsibilities after divorce or separation.

## 2.2. Flexible Working Strategies

Flexibility in the workplace is one element that is often related to the potential coping strategies of single parents in their attempt to balance work and family (Anttila and Oinas 2018). As mentioned above, the roots of a problematic work–life balance lie in the combination of financial strain and role strain. In terms of financial strain, the findings of Maldonado and Nieuwenhuis (2015) show that single parents, mothers especially, have a higher poverty risk, with the household demands increasing after separation. This poverty risk is higher for women because there is still a gender pay gap and women are more likely to have custody of the children after divorce (Arulampalam et al. 2007). However, increasing work time, which is one way to reduce financial strain, may increase the difficulties women face in maintaining the mothering role they want (Thielemans and Mortelmans 2019). As a result, single mothers have to find a balance between work and life that eases both the financial and role strain. Flexibility in the workplace may help them to find the time needed to invest in both roles (Van den Eynde et al. 2019).

However, flexibility in the labor market may entail a variety of measures that are not always oriented towards single parents and their needs. Van Gasse and Mortelmans (2016) developed a classification system in which they argued flexibility in the labor market may be beneficial to the employer, the employee, or to both. They described three ways to implement flexibility. First, structured flexibility options can reduce time and/or place constraints on a stable, well-implemented weekly basis. One specific example here is the use of flexible start and finishing times, with employers giving employees the option of arriving and leaving the workplace at various times (Friedman 2012).

Second, unstructured flexibility concerns the possibilities that employers have to rapidly change work routines. This kind of flexibility is often not employee-friendly but offers employers the opportunity to respond to sudden systemic changes in the market. One specific example is when employees are asked to work from home in the case of natural disasters or other events affecting the environment (Donnelly and Proctor-Thomson 2015).

Third, autonomous flexibility concerns the employee being able to define their work time and conditions. One specific example of this is work arrangements in which the employee has an office at work but can choose to work there or from home (Silim and Stirling 2014).

Van Gasse and Mortelmans (2016) also argued that flexibility can be categorized according to its focus on work time (flexitime), on workplace (flexiplace), or on the organization of work. While flexitime and flexiplace work arrangements might be of relevance to single parents, the organization of work is generally employer focused. Therefore, we will not examine the latter here. However, other approaches to work quality have looked more closely at the idea of flexibility in terms of the content of work (van der Kleij et al. 2013), resulting in the notion of flexible workplace behavior. From this perspective, not only can the organization of work be changed to meet the needs of single parent, but norms and attitudes towards work might also be adapted (Yadav et al. 2016). As Allen (2001) argued, the organization of work may create a family-friendly environment, but employees can also change their behavior within this context. Therefore, in our case, it is not only important to observe the context of the workplace, but also the behavior of single parents within it.

### 2.3. Differences in Parenthood Ideologies and Flexible Work Strategies

We began this article with the claim that finding a work–life balance can be more or less problematic for different people. We described how single parents struggle to find a work–life balance because they face a combination of role strain resulting from parenthood ideologies and financial strain as a result of the breakdown of the dual-earner family. We argued that flexibility at work may offer solutions. However, certain differences have to be taken into account in relation to this idea: They concern gender and socioeconomic status and thus imply an intersectional inequality (Özbilgin et al. 2011).

Firstly, both role strain and financial strain are worse for women than for men. Nieuwenhuis and Maldonado (2018) found that single mothers are among the most poverty-prone groups in European society. There is thus a greater need to increase work time for women than for men, which was also observed by Thielemans and Mortelmans (2019). Moreover, in terms of role strain, a gendered analysis by Vespa (2009) showed that women's parenthood ideologies are stricter and more family focused, while men's are more work focused. However, it should be noted that these gender differences are less present in egalitarian families than in non-egalitarian families. In terms of General Strain Theory, mothers' parental goals are therefore more difficult to achieve, unless a more egalitarian parental ideology is used by both partners (Agnew et al. 2002)

Second, there is a socio-economic gradient in terms of work–life balance (Kossek et al. 2015). While there are few studies that focus on the explicit relationship between socio-economic status and work–life balance, work environments that are associated with less balance are related to lower income groups (Bambra et al. 2008). Moreover, these economic sectors often use more employer-focused flexibility such as shift work, rather than employee-focused flexibility that could benefit single parents (Van Gasse and Mortelmans 2016). In terms of role strain, Taylor (2011) also argued that parenthood ideologies are not only influenced by gender, but also by class and ethnicity. As there exist intersectionality in between gender, class, ethnicity, socio-economic status, and parenthood ideologies, Özbilgin et al. (2011) argue there is a socio-economic gradient in work–life balance. Therefore, it can be argued that single parents with a lower socio-economic status will have greater role strain and consequently it will be more difficult to find a balance between work and family.

## 3. The Belgian Context

According to research of Bernardi et al. (2018), almost 8.5% of the Belgian households in 2010 were headed by a single parent. Many of these families are initiated by parental divorce. As we argued in the introduction, we look in particular to the divorce case as the transition to single parenthood after divorce or separation implies an increase in general strain (i.e., role strain and financial strain). Considering the divorce procedure, changes in the legal procedures of divorce have influenced the divorce statistics in Belgium and have an impact on the custody arrangements.

Since the 1990s, there has been an increase in divorce rates in Belgium due to a reform[2] in the legal system (Mortelmans et al. 2015). Regarding custody arrangements, shared custody is frequently used in Belgium. In Belgian courts, an exploration towards shared custody takes place before custody for one parent is investigated (Mortelmans et al. 2015). This means that shared custody is assumed first, before possible complications would arise.

Belgium is a country with an elaborate welfare system, with one of its aims being to reduce poverty (and financial strain) in single-parent families (Nieuwenhuis and Maldonado 2018). This means that there is a strong safety net for people who are confronted with adversity in the life course (Hemerijck and Marx 2010). One of these adversities, divorce has been on the rise since the 1990s, which means single parents are now one of the vulnerable groups that the social security system aims to assist (Mortelmans et al. 2011). To prevent single-parent poverty, a system of elaborate unemployment benefit combined with rather high amounts of child support and a secured minimum wage is used (Vandenbroucke and Vinck 2015). With these regulations, the minimum income of a single-parent family contains 1270.51 euros (excluding child support benefits). Child support is then calculated based on specific criteria (e.g., family income, special care needs, orphanage, etc.) and added to the minimal family income. Despite these measures, on average, 19% of Belgian–Flemish[3] families headed by a single parent have a poverty risk (i.e., have an income below the 60% average of the mean family income) according to Defever et al. (2013). As the policy measures do not always succeed in preventing poverty and poverty is usually situated in families with a lower working intensity, there is an awareness of poverty traps in Belgium, with the formal and informal support systems aiming to activate single parents to engage in the labor market (Raeymaeckers et al. 2008).

However, the opportunities available on the Belgian labor market differ for men and women. In 2002, an anti-discrimination law was introduced in Belgium (Adams 2003), designed to promote horizontal equity in different fields. It was, therefore, more generic than earlier laws on anti-racism and gender equity in the labor market. Although this resulted in a smaller pay gap between men and women, women's labor is still more often part-time than is men's labor (Marynissen et al. 2019). Women's labor is also overrepresented in sectors with lower earnings (Pillinger 2016). This affects Belgian society to a significant degree, with Mencarini and Sironi (2010) finding that the Belgian Global Gender Gap is greater than neighboring countries such as the Netherlands, Germany, and the United Kingdom.

## 4. Methods

This study was performed using a novel sampling approach, only previously used by Van Gasse and Mortelmans (2020a) and Van Gasse and Mortelmans (2020c) and a qualitative research approach. Considering the sample approach, two groups of participants were recruited. The first was a population of 13 single mothers who were recruited by snowball sampling, which began on social media. The second group was an elaborative population of 189 single-mother respondents who were recruited through the personal network of students involved in conducting the study. The primary population of 13 parents was interviewed using an unstructured, open, and in-depth approach, while the elaborative population was interviewed with semi-structured interview leads (based on the primary population interviews). Thus, each research population was approached differently. The use of structured interview leads by students allowed precoding in NVivo in separate data files (Mortelmans 2011). In this way, the unstructured primary data were kept separate from the data collected in the structured student interviews at all times, and they also served different purposes

---

[2] This reform implied there was no legal "guilt" question in a legal divorce process. The idea of proving guilt in court seems to have discouraged people from divorcing. The removal of the idea of guilt correlated with a significant increase in divorce numbers.

[3] There are large regional differences in poverty risks in Belgium.

in the analyses. All interviews were conducted in person by the main researcher (primary research population) or by a student (elaborative population).

All of the single parents were interviewed at a place of their liking.[4] Both samples included respondents who were the only adult in the household,[5] and who had at least one child younger than 18 (or between 18 and 24, if studying).[6] The child also had to live with the interviewee for at least 50% of the time. Therefore, single parents in shared custody arrangements were also included in the sample as long as they had childcare responsibilities. These criteria ensured that the single parents had to combine their parental responsibilities with their work routine. We interviewed people about their divorce and its aftermath retrospectively. Divorce or separation had taken place between 1 and 25 years prior to the interview, and we interviewed the participants in their role as mother and employee. Therefore, there was no explicit information on sexuality we could use in our analysis. However, due to the heterogeneous nature of the Belgian society, we might assume that there were heterosexual, as well as homo-, bi-, or pan-sexual respondents in our study population. Nevertheless, as we interviewed the respondents on their identity as a parent, there were no specific indicators to do a focused analysis on sexuality.

The double structure of data gathering took place in two different situations. The primary research population of 13 people was interviewed by the main researcher. The interviewees in the primary population were found through an announcement on social media and subsequently by snowball follow-up of different respondents.

The elaborative population of 189 single mothers were interviewed by students[7] involved in a qualitative methods course at the university. The students were evaluated on the quality of their contribution. An extracurricular workshop on interviewing was also organized by the first author. Working with student interviews entails quality risks, as students may commit fraud, deliver poor-quality interviews, or fail in some elementary aspects of interviewing and transcription. Therefore, the elaborative sample was controlled via sampled respondent calls, video data, closed coding similarity, and a respondent check. In total, 189 interviews were completed by the students, which allowed the triangulation and elaboration of the results.

The main theme of the interviews in both the primary sample and in the elaborative sample was the career trajectory of the respondents in their family contexts. To reconstruct the narratives of the interviewees, respondents were asked to draw timelines during the interviews to relate various events to each other (Kolar et al. 2015; Sheridan et al. 2011). In this way, a structured narrative could be reconstructed about the experience of divorce, its impact on work life, and the solutions single mothers found to establish a balanced situation.

The analysis was based on the principles of Grounded Theory, which was extended by focused comparisons within the elaborative population (Glaser 1998; Stern and Porr 2011). First, the classic approach in Grounded Theory analysis with open and axial coding was completed by the first author[8] on the transcribed interview responses of the 13 single parents in the primary research population. The transcribed interview responses of the 189 single parents in the elaborative group were precoded and classified by the students (Corbin and Strauss 2014; Corbin and Strauss 1990; Glaser and Strauss 1967).

---

[4]  Usually, this was the interviewee's home. Sometimes interviewees did not want to be interviewed at home so a different place was chosen.

[5]  This does not mean that all interviewees had not repartnered. It does, however, imply there was no cohabiting partner in the household with whom to share the household duties. There may also have been a significant other in the household in the past, but we did not particularly focus on these cases in the analysis.

[6]  It is common in Belgium for young adults to live with their parents while studying at university. We did not want to eliminate this type of single parent who has a student living at home, so we added this to our definition of single parents.

[7]  The students who participated in the data gathering were second-year Bachelor's students in the education programme in social sciences at the university where the authors work. The students were not allowed to interview single parents with whom they had close ties (such as their own kin).

[8]  Limiting the coding to the first author was done for practical reasons, as well as to ensure reliability in coding. However, the second author was involved in reviewing all of the coding procedures.

In this way, an initial theory was constructed by the first author following the more general rules of coding as described by Mortelmans (2017). This means that the initial interviews were structured with open codes. After this open coding, a similar sequence of coding appeared throughout the interviews. Subsequently, these sequences were coded in greater depth using axial coding to delineate these stages.

Following this coding, a first model was established, which we refer to now as the initial theory to look into the elaborative sample. This initial theory had the core structure of the grounded theory we present in this article. The theory was, however, expanded by using the elaborative research sample. The elaborative research sample was thematically coded and classified by students after they had been trained in NVivo. This was possible because the students had followed thematically structured interview guidelines. Furthermore, background data were gathered through a drop-off survey, which was processed using an extensive classification sheet (collecting 41 background characteristics per interviewee). This coding procedure made it possible for the researchers to later process the large number of interviews using coding Queries in NVivo.

Subsequently, the initial theory was tested and expanded through constant comparison within the elaborative research population, which now consisted of all the precoded student interviews merged in NVivo (Glaser 1998). By combining both structuring instruments (precoding and classification sheets), we used coding queries to perform focused comparison within the elaborative research population (Mortelmans 2011). This focused comparison with the elaborative sample enabled: (1) A deeper analysis of particular cases that were not, or only poorly, represented in the primary sample, (2) the expansion of the analysis of respondents with specific characteristics that were not available in the primary sample, and (3) the triangulation of the results in the search for negative examples or more nuance in the analysis. The elaborative research sample thus contributed to the results by deepening and expanding the analysis and offering a reliability support due to its triangulation power.

## 5. Results

Based on the interviews, we identified four different perspectives on the combination of mothering and work taken by single mothers. Although the typology we present mainly reflects a continuum of circumstances people may find themselves in, we were able to distinguish mothers who had either a 'lean' motherhood ideology or a 'strict' motherhood ideology, and mothers who looked for flexible workplace options or those who remained in rigid workplaces. Moreover, these circumstances and roles were not always chosen by the single mothers; some were trapped in a situation that governed their actions. To introduce the single-parent perspectives on motherhood and the workplace, below we first explore the motherhood ideologies and flexibility options in the workplace that we identified. Subsequently, we introduce the four perspectives on motherhood and work that single mothers may have.

### 5.1. Lean vs Strict Motherhood Ideologies

The first axis of our typology consists of the opposition between lean and strict motherhood ideologies. A motherhood ideology can be seen as a set of norms and values regarding motherhood. Within this set of norms and values, women internalize a mothering role to which they are committed. However, how demanding and restrictive an internalized motherhood ideology is can vary between individuals.

In lean motherhood, this set of norms and values is interpreted in a flexible way. Moreover, the mothers believe that the roles they take up are adaptable. At the opposite end, a strict motherhood ideology is a very rigid set of norms and values, where mothers consider their role to be determined and little adaptable to a changing environment. These mothers talk about the internal obligation to engage in practices that are necessary if they are to be 'a good parent'. This results in very rigid role patterns that they believe they should meet.

Thus, the level of flexibility of a motherhood ideology has consequences in terms of values regarding motherhood and in terms of the roles and functions mothers assign themselves as parents.

In summary, a lean motherhood ideology provides an individual compass of morals and values that functions as a guide to a single parent in their parental tasks.

> *'Good motherhood, for me is [hesitates] mainly vague things. Being there for your children. Giving them a warm nest feeling, the feeling that you can be there for them. It's not really about doing things but about generating this feeling of warmth.' (Laurence, 38, single for 11 years)*

Although a strict motherhood ideology may be based on the same principles as the lean version, it functions as a more structured and rigid set of rules guiding a mother's commitment. If one of these internalized 'checklist items' is not realized, it may feel like failure and cause distress. The internalized stress about good parenthood is thus higher when the motherhood ideology is more strict and rigid.

However, it is important to note here that there is not an opposition between strict and lean ideologies but rather a continuum. Along this continuum, a lean motherhood ideology can be thickened with extra obligations that make the ideology stricter, and vice versa. Thus, the mothers could be distinguished in the rigidness of their motherhood ideologies, but this was a continuum, with few people on the extremes and most somewhere in between.

> *'I am a terrible mother, yes [ … ] That's my experience. [Child name] has a much better daddy than most kids, because he plans activities with his daughter and I don't have time to plan activities. I invite her friends to come and play with her because it's easy for me … I can do my own thing then. I want to use my time for her, but I just can't do it the way her father does.' (Sybille, 40, single for 1 year)*

*5.2. Flexible Workplace Behavior vs, Rigid Workplace Behavior*

The second axis concerns the difference between flexibility and rigidity in the workplace. We define the concept of flexibility as the level of adaptability of work that takes family life into account. Rigidity, by contrast, is the inability to make adjustments in the work environment to improve the work–life balance. We have also broadened the idea of flexibility and rigidity at work beyond practical organization, as employees in a flexible organization might still experience a restricted sense of commitment, while other employees in more rigid organizations may have a stronger sense of the delineation of work and family. In other words, flexibility or rigidity at work have an organizational aspect and an internal aspect concerning the orientation of employees towards the workplace.

The first aspect of flexibility in the workplace thus concerns the organization of labor. In these concerns, we rely on the same terminology and distinctions as Van Gasse and Mortelmans (2016). Single-parent employees can either choose *flexitime* contracts, focusing on the time and hours they have to work, or *flexiplace* contracts, adapting the spatial conditions of work. People using these strategies do not change the work itself but change the rigid delineation of work in terms of time and/or place. These strategies are very much aimed at problem-solving and mainly used to counter the challenges of *time*, *place, and financial constraints*, while maintaining previous work standards. The use of flexitime and flexiplace contracts can help to enable a high level of commitment to the family as well as the workplace.

> *'I changed work so I had less fixed hours, like a nine to five job, but a job where it was possible to start later or leave early. The only thing that was needed was someone doing the job all the time. There is solidarity among colleagues, so no one is bothered if I arrive later and have to leave to get my children from school' (Mathilda, 46, single for 7 years)*

The second aspect of flexibility in the workplace goes beyond the practical organization of work. It concerns the internal norms and values someone has towards their work practices. This value framework is partially influenced by the culture of a workplace but also by the individual's attitude to work. If people are more work-oriented and have a higher commitment to work, they will attempt to do their best at the job at all costs. However, someone with a more flexible stance will bend their values according to the situation in which they find themselves. This may mean that people lower their aspirations, but also that they change the way they approach their tasks.

*'Before the divorce, I was doing my job all the time. I did extra hours ... every week! I was at the
office early in the morning and I stayed really late. There was always someone with the children so I
could do it, but I had to stop this after the divorce. I just do the hours I'm payed for, not trying to do
things better than my colleagues, I'm gone at half past 3. I have a very stable balance and you know
what? The earth is still turning the same lame circles.'. (Valery, 38, single for 2 years)*

Flexibility at work is thus not only about taking advantage of the flexible organization of work,
but also about how single parents internally negotiate the boundaries of their work activity. In some
instances, employees in a flexible work environment still maintain strict and rigid work values and
commitment. In other cases, single parents who are in very strict and rigid organizations can find
creative and flexible ways to engage in this structured environment. Therefore, flexibility at work
should be seen as determined by a combination of the possibilities that are presented by the employer
and the individual's internal attitude towards work.

*5.3. Single Mothers' Perspectives on Motherhood and The Workplace*

Based on the two aspects of (i) flexibility/rigidity, which we identified in the single mothers'
descriptions of the adjustments they made in their working careers and (ii) the distinction between
lean and strict motherhood ideologies, we developed a two-dimensional typology, which generated
four motherhood–workplace negotiation perspectives. These motherhood–workplace negotiation
perspectives were labelled: (i) Reinvented motherhood, (ii) work–family symbiosis, (iii) work-centered
motherhood, and (iv) work–family conflicted motherhood. As is the case for all typologies, it is
important to note that this is a heuristic instrument, which is designed to offer us a better understanding
of single mothers' career behavior, rather than being understood as a definitive categorization of all
single mothers. A visual presentation of these perspectives can be consulted in Figure 1.

**Figure 1.** Perspectives on the combination of motherhood and work.

5.3.1. Re-invented Single Motherhood

In *re-invented single motherhood*, we find single mothers who have a lean motherhood ideology
and flexible workplace behavior. This means that they find a satisfactory work–life balance through a
flexible work environment and a flexible motherhood ideology, which results in a new perspective
on motherhood.

*'When I was single, I changed my mindset. I was always struggling with those thoughts ... I am not
able to be a good mother, or a good employee [ ... ] I was a mess, an underachiever in all areas of my
life ... Until I realized that I may not have been the mother that I wanted to be at first, different things
were important. I also didn't have the career I dreamt of when I was studying, but I was doing my job
in my own way' (Stephanie, 37, single for 7 years)*

Firstly, in reinvented motherhood, mothers use a lean motherhood ideology. Although some mothers changed their motherhood ideologies after divorce, this was not the case for all single mothers. Some mothers already had a lean motherhood ideology that could be interpreted in a flexible way and stopped them from pursuing unattainable goals in work and family life. A lean model of norms and values regarding parenthood can be more easily adjusted to the changing demands of single motherhood. However, some single mothers in this category may not have started here but had to cut through their stricter parental values and motherhood ideology to develop a leaner model.

Secondly, in reinvented motherhood, mothers exhibit flexibility in their workplace behavior. Either they look for a job with a more flexible organization of labor or they change their attitude about work, or both. In some cases, the single mothers already had a flexible and family-friendly workplace that ensured they did not experience insurmountable difficulties after divorce. However, other single mothers in this category had to look for or create a more flexible and/or family-friendly workplace and succeeded. This implies that the single mother had agency, being able to look for new work or negotiate with their employer to adjust the work environment to their changing needs. This agency, however, is always limited to the context a single-parent employee is in.

### 5.3.2. Work–Family Symbiosis

The second perspective on motherhood and work that we identified is *work–family symbiosis*. This position is characterized by a family-centered work ethic. Rather than developing a leaner motherhood ideology, single mothers in this category look for ways to bend their workplace behavior around a rigid motherhood ideology, which they remain committed to after divorce.

> *'Mothering is about looking for balance. In the first place, you have your responsibilities to your child [ . . . ] your priority as a mother is to give them a good, consistent upbringing that leads your child to become a good human being. [ . . . ]. Nurturing is also about putting food on the table but a good boss understands your priorities as well. Neither employers nor parents should ever prioritise work over children' (Britt, 55, single for 12 years)*

This position may also result in more critical behavior if a single mother decides to look for new career opportunities (although not all single parents would risk changing jobs after divorce). A job has to be, in the first place, family-friendly in its organization, while the job content seems to be less important, as long as it is possible to combine it with a stricter motherhood ideology. Single mothers who have a work–family symbiosis perspective thus look for jobs that offer a flexibility that enables them to give up few of their motherhood aspirations.

> *'I was looking for a new job, but if I couldn't negotiate the working hours, or the possibility of working from home, even if the job was as interesting as it could be, I wouldn't take it.' (Sabine, 44, single for 6 years)*

The combination of this strict motherhood ideology and flexible workplace behavior results in a family-centered lifestyle in which everything is a function of parenting. Therefore, single mothers who opt for a work–family symbiosis also seem to alter their career aspirations. One remark that was made many times was that having a career was no longer something to which they could aspire.

> *'I know they give promotions to these young – let me call them naïve – people, who don't know their priorities yet, or who don't know what the toll of these positions will be on their later life as a parent.' (Sylvia, 49, single for 13 years)*

### 5.3.3. Work-Centered Motherhood

The third perspective we identified is *work-centered motherhood*. Unlike work–family symbiosis, the single mothers in this category have a work-centered ethic. This results in single mothers who meet the rigid demands of their workplace, while looking for childcare solutions outside their primary

family system. Some of the single women in this group recounted how a failed marriage led to solace in their other life project—their career.

> 'When I got divorced, I locked myself up somewhat at work. I worked hard and the appreciation at work felt so good and my parents took care of the children. Sometimes I felt bad because I was so absent [ . . . ] but then again, I thought, if they want food on the table, mummy needs to work and if they want a happy mummy, mummy has to work.' (Jane, 38, single for 6 years)

Although there are some narratives like the one above that suggest some 'escapism' through work after divorce, and in which work is used as the ultimate form of self-development in life, more often the narratives in this group suggested that the single mother was simply trapped in a rigid work system. However, these mothers adjusted their motherhood ideology towards a slimmer, 'lean' motherhood ideology to deal with the high demands of the workplace. Therefore, we can distinguish a kind of *forced* lean motherhood ideology from one that is *chosen*, with not everyone voluntarily adjusting their motherhood ideology.

> 'I had asked [them] to let me try, I said I could do it as a full-timer, I would stay longer and take no breaks, not even a lunch break, I needed the money. I also didn't want to do a boring part-time job, and I assumed I would get quicker in my routines. Reality proved that I just gained less time for my children, but I had to do it.' (Jinthe, 52, single for 17 years)

### 5.3.4. Work–Family Conflicted Motherhood

The final perspective we could distinguish is *work–family conflicted motherhood*. In this category, single mothers want to retain a strict motherhood ideology while still meeting the demands of a rigid workplace. This can result in a problematic situation, in which the two life spheres are in conflict.

> 'It should be 50-50, but I feel that I have run out of energy when I'm home and I can't be a good mother because of that [ . . . ] It's very busy and Dutch(Allen and Hawkins is not my mother tongue so I have to work hard to keep the job [ . . . ] and I can't do the things I want to do to be the mother I want to be.') (Amina, 43, single for 7 years)

This perspective is often the result of not being able to adjust workplace behavior and/or not being willing to relax one's motherhood ideology. Flexible workplace behavior is not open to everyone and not applicable in many contexts. Mothers who find themselves in this category are thus often family-centered mothers who are simply not able to adjust their work environment in accordance with their strict motherhood ideology and who do not want to compromise this ideology for their work.

Another narrative in the work–family conflicted motherhood category is the fear of change in the work environment and/or fear of the consequences of adjusting their motherhood ideology. This is also a reason for single parents to stay in a work environment that is incompatible with the aspirations they have as a parent. As career changes imply insecurities, some parents choose the security of a job they know, although this is incompatible with their new family life situation.

### 5.4. An Economic Gradient in the Perspectives of Single Parents

It should be noted that for many single mothers, the motherhood perspectives described above are not voluntary choices. The opportunity to adapt one's workplace behavior to family needs is generally only available to specific groups of people (e.g., higher SES and/or certain employment sectors). Therefore, it can be argued that there is an underlying issue of inequality in the options that single mothers have in relation to adjusting their motherhood ideologies and workplace behavior. An economic gradient in the perspectives on motherhood and work can be assumed based on the narratives of those single mothers who did not have access to flexible work options. The literature also shows that parenthood ideologies differ across socioeconomic status (Taylor 2011).

*'Some of my friends can work from home and it would be such a relief to be able to do something like that, even for one day a week, but it is simply impossible in a job like mine'. (Angelina, 43, single for 6 years)*

Moreover, as argued, the conflict faced by single mothers involves a combination of role strain and financial strain. In the four perspectives we presented above, single mothers release the pressure of one or both by taking a more flexible organizational or normative framework in relation to work or their motherhood ideology. However, for some single parents, the financial strain and job insecurity are so great that they are not in a position to demand the more flexible organization of work and/or loosen their normative framework, due to fear of losing their job. In these cases, a single mother's socioeconomic position can clearly limit the perspective she can take in dealing with both strains.

*'I knew I had to maintain the efforts I made at work to not be fired and that primed for me. There was no time to cry, no time to lose and the last thing you want is to be jobless in such a situation.' (Marthe, 27, single for 2 years).*

Thus, for all these reasons, rather than a definitive typology of perspectives that single mothers can take, our typology must be interpreted as volatile and selective. Not all the options are open in every case, and some people may face a greater amount of risk when attempting to establish a more comfortable position.

## 6. Discussion

The four perspectives that single mothers may have on the work–life balance presented in this study should be seen as an element of the transition that they make after divorce and/or separation. During this transition, single mothers experience high stress levels that ultimately drop when lives are reorganized, and a new balance has been found (Booth and Amato 1991). This reorganization partially takes place in the household, when single parents restore balance in the way they organize their household, and also partially in the work context (Van Gasse and Mortelmans 2020a). Single mothers are confronted with many challenges regarding the combination of work and family life. The work environment often fails to be family-friendly, while the aspirations of mothers in their parenthood role often demand a high investment. Therefore, single mothers are prone to find themselves in a conflicted work–family situation (Van den Eynde et al. 2019). Single parents can take different perspectives on this issue. In this article, we looked into the perspectives single mothers may take while staying active in the labor market, while acknowledging that dropping out may also be a consequence of divorce (Eamon and Wu 2011).

The position of single mothers in the workplace can be seen from two perspectives. From a financial strain perspective, after separation, single mothers hope to increase their work activity to support their family on their own (Thielemans and Mortelmans 2019). It should also be remembered that single mothers are more at risk of poverty, since they are more likely to have custody and suffer gender inequality in the labor market (Fuwa 2004; Sodermans et al. 2013). However, after divorce, there is also an essential time period in which individuals grow into a new 'single' parent role (Taylor 2011). In this role, single parents must evaluate their parental role, which causes *role strain* (Nomaguchi and Milkie 2017). In the context of single motherhood, this role strain can be interpreted from the gendered institution perspective (Killewald 2016). From this perspective, the role frameworks with which people comply are often gendered and more demanding for women. The parental roles of mother and father, for example, are partially socially constructed patterns that reflect culturally interpreted gender differences (LaRossa and Sinha 2006). For mothers, this role is not only constructed by societal expectations but also by individual mechanisms, such as aspirations and maternal gatekeeping, which defines the boundaries of their motherhood identity (Puhlman and Pasley 2017; Salmela-Aro et al. 2000). The socially constructed role parents are thus partially internalized within their motherhood ideology. Therefore, the single mothers' post-divorce work–life perspectives

presented above should be seen as combining the economic *financial strain* perspective and the role-oriented *gendered institution* perspective (Killewald 2016).

In the typology we described, it seems desirable to move single mothers away from the conflicted work–family position, but that this is not always a simple option. Firstly, gender differences make the strain problem greater for women, with fatherhood ideologies generally less demanding (Glauber and Gozjolko 2011). Secondly, socioeconomic status also affects the perspectives that single mothers can take in relation to the labor market. In relation to the latter, Taylor (2011) argues that single mothers with a lower SES generally have more rigid motherhood ideologies, and generally have fewer opportunities to adjust their working hours (see also Hari 2017). Therefore, Single mothers with a lower SES are more likely to be caught in a work–family conflicted motherhood perspective. This assumption has also been confirmed by Van den Eynde et al. (2019) and resonates with our findings. Due to the qualitative nature of our research, we could not present this socio-economic gradient on a representative sample. Nevertheless, the relative high amount of cases we studied can serve as a strong indication for more role and financial strain in groups with a lower socio-economic status that has to be taken into account. Moreover, we can argue that this strain is an accumulated strain caused by stressors on many domains and can therefore be attributed to intersectional differences in between groups of single parents (Özbilgin et al. 2011).

## 7. Conclusions

In this study, we described the work–life perspectives of single mothers, developing a typology of the various ways in which single mothers cope with the potential conflict between work and family. Rather than focusing on the financial strain perspective—looking solely at the difficulties single mothers face in increasing their work time to avoid poverty—we considered the financial strain perspective in combination with role strain in a society understood as a gendered institution (Killewald 2016; Thielemans and Mortelmans 2019). We conceived of the conflict between work and family as a conflict between two systems in which single mothers play roles that need to be balanced. This restoration of balance can be found through greater flexibility in motherhood ideology and/or greater flexibility in workplace behavior. However, not all of the options are open to all single mothers.

We contributed to the theoretical literature by introducing the work–life conflict in a General Strain Perspective (Agnew et al. 2002). Moreover, the work–life conflict that is experienced by single parents can be seen as a result of the cumulated strains in different life domains. In this analysis, we focused solely on the role strain regarding the parental role and financial strain at the workplace. The flexibility that is sought by single parents illustrates that single parents try to relieve strain by looking for flexibility in many life domains.

These findings have implications for practitioners, employers, and policymakers. Regarding practitioners and people who work with single mothers after divorce, it is important to focus on motherhood ideologies. When people are reorganizing their lives after divorce, it is not only important to look at changes in household routines, but also at their role aspirations as a mother (Van Gasse and Mortelmans 2020a, 2020b, 2020c). Motherhood ideologies partially reflect the motherhood aspirations of single mothers, and this is also an important aspect in their transition to single parenthood (Taylor 2011). Single mothers not only face psychological challenges, such as feelings of failure after a marriage break up, but also the practical need to maintain their mothering efforts on their own. Therefore, practitioners could play a valuable role in supporting single mothers by exploring what they find important in motherhood, and thus help them focus on the values they hold essential in their maternal role.

In this respect, practitioners might help single mothers think through how a very strict motherhood ideology might be relaxed, allowing more flexibility to cope with the challenges of single parenthood and reducing the stress related to role strain (Booth and Amato 1991). It should be noted that the great demands of motherhood are partially socially constructed and should thus also be looked at in their cultural contexts (LaRossa and Sinha 2006).

With respect to employers, it is important to construct a family-friendly workplace (Strachan and Burgess 1998). The standard workplace, with rigid work hours and little task flexibility, presents difficulties for single mothers, who face the pressing needs of combining the life spheres of work and family in a more intense way. However, not all forms of flexibility are beneficial to single-parent employees (Van Gasse and Mortelmans 2016). When focused on the needs of an employee, a good degree of time, place, and task flexibility might help employees who are challenged by the rigidity of standard workplaces. Such flexibility might even be focused on risk groups such as single parents, who are willing to work but are short on time. However, it should be noted that flexibility not only concerns the organization of work, but also the value framework of the employee. Although this is an individual employee trait, a workplace culture can help to enable greater flexibility in the employee's attitude to address the issue of role strain (Varpio et al. 2018).

Finally, in relation to policy, the general inequalities in the labor market should be acknowledged. Policy makers should thus both address financial strain of single parents as their role strain. First, in the context of financial strain, the Belgian minimum income and child support benefits are ways to reduce single-parent poverty (Vandenbroucke and Vinck 2015). Moreover, focused targeting on vulnerable groups decreases the Mathew effect associated with general child support. Second, policy makers can reduce role strain by supporting employers to apply family friendly work organization. Awareness has to be risen that not all single mothers are in a position to gain access to more flexible work agreements. As there are strong indicators for a socioeconomic gradient in the opportunities available to single parents, a Mathew effect is implied. This is an intersectional inequality related to family status, gender, ethnicity, and social class. Social policies should thus identify these differences to better tailor the measures they implement. Therefore, focused policies that target single parents and ensure flexibility for single parents employed in more rigid work sectors might help improve the quality of working life of single mothers (Brady and Burroway 2012). To reduce strain, policy makers can choose to enable single parents more to adapt structured flexible work arrangements (Van Gasse and Mortelmans 2016). Therefore, single parents should be targeted as a vulnerable group regarding rights to request flexibility (Mairhuber et al. 2009). Targeted flexibility may reduce strain as it enables single parents to reorganize their work to their parenthood ideals.

*Limitations and Prospects for Future Research*

There are some limitations of the study that have to be addressed at the end of this article. However, these limitations can also be seen as prospects for future researchers to expand and adjust the theory that we presented.

First, we chose to study work–life conflict as an interplay in between work and family life. However, it has to be noted that work and the family are not the only domains of self-development for single parents. Therefore, the abstraction of work–life conflict to a strain conflict in between financial strain and parental role strain ignores the importance of other life domains. Bakker and Karsten (2013) for example argue that also leisure time is also an important area to address work–life balance to that we do not take into account in this study. Whereas the simplification of work–life conflict to a two-strain problem, adding other strains (e.g., leisure strain) can expand the theory on different domains, enhancing the idea of perspectives towards work and motherhood to new perspectives.

Second, we analyzed the data without a particular focus to elements such as ethnicity, class, and sexuality. There are, however, strong indicators to make a contribution towards research on intersectionality in our research, in which these elements can prove to be valuable key factors. Moreover, the socio-economic gradient can be a gradient that runs over multiple domains.

Third, whereas the qualitative nature of our research enabled us to explore motherhood ideologies at the workplace in depth, it does not support a strong claim for the socio-economic gradient we observed. Future research should confirm the strong indicators we observed regarding a socio-economic gradient on the position of single mothers regarding their work–life combination.

Fourth, there is a time component that should be addressed in future research. In our retrospective interviews, we added cases of single motherhood with mothers that recently transitioned to single parenthood and single parents that made transitions a longer time ago. Therefore, we described single parents that were well-adjusted to their life as a single parent as well as single parents struggling with the initial post-divorce challenges. We might assume that there would be a process of adaption in the transition process, but longitudinal data is needed to study the adaption single mothers make in their position regarding motherhood and the workplace.

**Author Contributions:** Conceptualization, D.V.G.; Formal analysis, D.V.G.; Methodology, D.V.G.; Resources, D.M.; Supervision, D.M.; Writing—original draft, D.V.G.; Writing—review & editing, D.V.G. and D.M. All authors have read and agreed to the published version of the manuscript.

**Funding:** This research was funded by FONDS WETENSCHAPPELIJK ONDERZOEK, grant number 140069.

**Conflicts of Interest:** The authors declare no conflict of interest.

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
