# Peer review of "Single Mothers’ Perspectives on the Combination of Motherhood and Work"

_socsci, doi:10.3390/socsci9050085_

Round 1

Reviewer 1 Report

Overall: This is an interesting paper on an important topic. There is a lot of potential to make a strong theoretical contribution to a few related literatures. Below I offer feedback intended with the goal of helping the authors strengthen the paper and attract readers. I hope the authors take the feedback in that spirit.

Areas of Strengths:

  • Strong initial frame to the paper
  • This is an area that needs additional research and I think that interviews are a good way to get at these experiences and examine processes that transcend national boundaries.
  • Well defined research question, with a clear rationale as to why single mothers were their focus. Consider discussing why inly discussing divorced women rather than separated or women who were not married to their partners.
  • The discussion of maternal gatekeeping was interesting, and served to contextualize ways women reinforce boundaries
  • I think the inclusion of students in this process is admirable, and I think the cross-checks to assure usability of the interviews was well considered.
  • I find the themes of flexibility and rigidity for mothering ideology and within the workplace and the perspectives on the combination of work and motherhood to be well discussed and represented within the article. This is an interesting contribution to the larger literature and can open a new avenue of discussion particularly in respects to mother’s parental self-efficacy and the relationships between work and mothering.

Areas for improvement:

  • The authors discuss using grounded theory, but honestly very few researchers truly take a grounded theory approach. It is perfectly reasonable to take an inductive approach, but to adopt smart methods tools from grounded theory.
  • Somewhat related, the authors repeatedly discuss strain’s impact on behavior. I strongly suggest the review Robert Agnew’s General Strain theory (GST) (1992, 2006) to better inform readers and also to better explain how financial strain is impacting these women. GST, while usually associated with crime, is increasingly used to explain non-criminal coping particularly for women who typically respond to strain differently than men.
  • This study was conducted in Belgium, a country with welfare system and labor laws are likely less well known to a broad range of readers. While the author does briefly discuss these, more detail would strengthen the paper and help guide readers. For instance, I am still unclear as to how much support there is for those in poverty? This is important to address within the larger research as it impacts how much economic strain is experienced by single mothers. So a more thorough explanation of this is needed. The location of the research should also be specified in the abstract and introduction to make the reader aware. It will help the flow of the paper, which seems to make a swift turn from general to specific quite quickly.
  • Intersectionality needs to be addressed in this paper. The authors focus on gender, but only mention that parenting ideology is influenced by race and class (citing Taylor 2011). However, the authors do not discuss the role of race, sexuality, education, social networks, and state supports services further. At the very least, race and class should be addressed more substantively in the paper. Even if the authors are only looking at white middle-class divorced women, gender, race, class, sexuality, et cetera are all still relevant.
  • The authors continually discuss single motherhood as being a mainly a result of divorce, but do not discuss if this is representative of single motherhood in Belgium other European.
  • Related, the authors do not address non-heterosexual relationships that result in children, which seems to reflect an assumption of heterosexuality and marriage throughout the literature review. This assumption needs to be addressed.
  • As written, the authors seem to assume in the literature review that single (divorced) mothers are 100% in charge of their children, with no involvement by another parent. Are there not joint custody arrangements or visitation in Belgium? Do other parents, as a rule, sever contacts with their children upon divorce? What are the rates of single fatherhood? This should be clarified.
  • The authors seem to make the assumption that mothers only have work and mother identities/selves when it comes to work-life balance but it makes no allowances for the need and right for a separate human person identity independent of work or children. This is a luxury afforded fathers, but not mothers, and should be addressed when talking about work-life balance.
  • The discussion and conclusion sections are good, and the authors really engage the literature well. I think the discussion and/or conclusion is a great place for the authors to address the state social supports in Belgium, especially as they compare to other nations, as it could impact the perspectives of the women in their sample. This would also really situate the sample into a larger context. They should also address any limitations that the article has. Most importantly, the authors should make a strong case for their theoretical contribution to the literature.

Small things:

  • If the authors are going to even mention “doing gender” (West and Zimmerman 1987) as they have, they really should cite the authors of that theory.

Author Response

Dear reviewer,

We like to thank you for the elaborate review and appreciation for the study as was presented in its initial form. You gave us also a series of remarks to strengthen the paper even further and attract more readers, which we addressed with care.

A first comment considered the use of Grounded Theory. We rephrased our methodology section to take your concerns into account. although the analysis took place in the Grounded Theory research tradition, the description of Grouded Theory by Glaser and Strauss, and more specifically Corbin and Strauss is not entirely suited for the large amount of interviews we used. Therefore, we rephrased it that especially the initial theory was largely based on the Grounded Theory approach, and expanded with focused comparisons in the elaborate sample. Therefore, our analysis is inspired by Grounded Theory rather than an exact use of the classic description. 

A second comment suggested to include Robert Agnew’s General Strain Theory. This is an interesting theory, although it is definitely more related to criminology. We decided to include General Strain Theory to describe what strain exactly implies. We think that General Strain Theory opens a good perspective to describe how life stressors affect human experiences and behaviour. Refering to General Strain Theory might inspire future research to adopt General Strain Theory to other life domains. 

A third concern was about the Belgian context. We understand that a proper description of the Belgian case increases the study’s attraction to readers from other countries. Therefore, we added a short description on the Belgian measures to prevent single parent poverty in the description of the Belgian context and added Belgium in the abstract and introduction to clarify that the reported results are derived in a Belgian context.

A fourth comment stressed the importance of adding intersectionality to the paper. We think there was already an implicit argument towards intersectionality in the paper when discussing the differences in parenthood ideologies and work-life balance. We clarified this by adding a couple of sentences explicitly addressing intersectionality in the literature as well as in the discussion. We also added this to the limitations as we think future research should address this issue more in focus.

Fifth, it remained unclear why we chose to focus on single parents due to divorce or separation. We added an argument why we chose for single motherhood after divorce. The main reasoning behind this choice, next to the availability of the sample, is the reported stress levels during the reorganization process after divorce, illustrating the role strain related to the divorce process. We think that widowhood and/or single motherhood by choice (which are other pathways leading to single parenthood) deserve focused attention in different studies. Also regarding the assumption of marriage, we tend to use divorce and separation as a way to both study former wedded couples and former cohabiting couples. We added this in a footnote for the reader and in the limitations as well

A sixth comment was related to the fifth and was concerned about the heterogeneity of the group of single mothers. It remained unclear whether we included only heterosexual couples. Therefore we adjusted our manuscript. Even though we did not interview respondents based on their sexuality and sexuality was not a topic of the research, we did not include only heterosexual single mothers. Therefore we added this statement in the method section:

“We interviewed the participants in their role as mother and employee. Therefore, there was no explicit information on sexuality we could use in our analysis. However, due to the heterogeneous nature of the Belgian society, we might assume that there were heterosexual, as well as homo-, bi- or pan-sexual respondents in our study population. Nevertheless, as we interviewed the respondents on their identity as a parent, there were no specific indicators to do a focused analysis on sexuality.”

The seventh comment made us aware that it was not clear what we did with shared custody and how custody arrangements are made in Belgium. We are aware that this was not clear in the initial manuscript. Therefore, we added a sentence in the method section that clarified that we did include parents in shared custody arrangements as long as they had at least custody for 50% of the time. We assumed this was needed as indicators for the work-life stressors of single parenthood. We also added some more details on the divorce procedure in the paragraph on the Belgian context. As we focus specifically on single mothers, single fatherhood is rather a phenomenon in the margin. Therefore, we did not elaborate too much on the topic of single fatherhood. Nevertheless, the description of the Belgian custody arrangements indicates that many single parent-families are father-headed as well.

The eighth comment pointed at the narrow definition of work-life balance we used. Presenting the work-life conflict as a role strain/financial strain problem, we seem to neglect other area’s in life. We agree work-life balance is more than the abstract conflict we constructed in between parental role and the financial strain. Therefore, we added this to the limitation section and referenced to consult other research of e.g. Bakker and Karsten (2013) who refer to work, care and leisure. For the purpose of this study, the abstraction of other life domains (e.g. leisure) enabled us to perform an in depth analysis on the conflicted motherhood ideologies, but we are aware that this partially ignores other areas of life that are important in any person’s life.

The ninth comment supported us to make stronger claims regarding social support measures and to address our theoretical contributions more firmly. Regarding this comment, we elaborated our conclusion with a small discussion about policy measures that can target single parents both to reduce financial strain as role strain. We do think however that our study is not the place to produce strong statements about policies as we did not study social policies in particular. Therefore, we think future research can focus on cross-country comparisons on work-life balance of single parents, studying policy measures in particular. We did, however, include a careful contribution to policy measures that can be used to reduce financial and role strain.

Next to the nine major remarks, reviewer 1 had a small remark to add West and Zimmerman (1987) to the reference list when we discuss doing gender which we did in the new manuscript.

We hope that we adjusted the manuscript well and improved the article following the guiding comments we were given.

Reviewer 2 Report

Dear Authors!

This is a valuable contribution addressing a problem of high relevance.

However, there are some minor issues that could/should be improved, or at least clarified, to make the interpretation more evident. 

1. Indeed, work-life balance is, or more precisely, it was a gendered issue. More and more studies, especially from the last ten years address the issue of fathers having huge problems with WLB. This study focuses on single mothers, therefore it should indeed not discuss fathers in more details. However, it is worth mentioning that the issue of work-life balance of men is increasingly coming in the focus of attention. 

2. The heuristic instrument of motherhood-workplace negotiation perspectives is a key in the analysis and indeed a valuable typology. From the presentation of results it does not become clear what the added value of the double structure of data collection (primary and elaborative population) was. For instance, did the two motherhood ideologies - the lean one and the strict one - emerge from the interview analysis in the primary research population (the 13 interviews), and then the elaborative interviews were analysed with regard to this typology? I understand yes, however, this aspect should be more pointed out more clearly.

3. The negotiation work (adjustment, restoration of balance) of single mothers is seen as an element of transition. There are some women who have been divorced for over 10 years, which implies that this transition had taken place a long time ago and must be over now. Then, transitional times are dealt with in a retrospective view, which might make a methodological difference compared to those who have divorced for one or two years and who are, indeed, still in transition and in the very middle of the adjustment process. I suggest this point to be made clear in the discussion.

4. There is an economic gradient in the chances of single mothers to adjust their workplace behaviour to their family needs. This is a crucial point. The authors refer to the differences in parenthood ideologies across socioeconomic status (Taylor, 2011), without, however, further addressing this issue. I very much miss the statement resulting from the interview analysis that single mothers with low SES rather take the work-family conflicted motherhood perspective. At the end of the Discussion part, this point is too carefully made as an assumption. However, from this numerous sample size, although not representative, such an inference could and should be made with more self-confidence. Also, at the very end, authors argue that "there may be a socioeconomic gradient in the opportunities available to single parents". But there IS one, and it was stated before in the results section! I recommend somewhat more assertiveness in declaring these important results.

5. In the introduction, the authors claim that preceding divorce, women spend more time at work, a phenomenon that is explained by anticipation, precedent and selective antecedent. However, we only understand the view on anticipation, this one is shortly described, and the reader is left without a hint as to what the other two concepts imply. Not much is needed but two or three sentences to clarify this.

Last, two minor issues:

a.) Maternal gatekeeping is mentioned in the literature review and in the discussion, but are by no means relevant for the results. I suggest dropping this line at least in the discussion part.

b.) It is a little bit incoherent in one interview why the lack of Dutch knowledge is a disadvantage at the workplace for a Belgian mother? There should be a hint enlightening this question, which is not obvious for the reader. 

I value the work done with the paper. I am sure that just a little bit of clarifying work will make this contribution much more understandable, traceable and it will increase its heuristic value for the reader.

Author Response

Dear reviewer,

We like to thank you for the appreciation of our study and the practical and concise remarks to improve our manuscript. We will pointwise discuss the improvements we made regarding your concerns.

A first comment stated that we should not neglect the issues in work-life balance of single fathers but we shouldn’t discuss this issue in too much depth. We understand that it is important to mention both genders. We added a couple of sentences in the introduction, pointing to the literature on fatherhood but we also argued why the issues of fathers are essentially different than those for mothers. This is due to the different shift that fathers make from breadwinner to breadwinner/caretaker and mothers from caretaker to caretaker/breadwinner. For this reason, we believe it is important to use focused studies to investigate the perspectives in depth.

A second comment was a methodological concern. You pointed out that it was not clear what the use of the elaborative sample was but the interpretation you had was correct. We clarified this in the method section. The elaborative sample serves to deepen and elaborate our findings and to increase the reliability of our findings through focused triangulation

A third concern regarded the issue of transitions, as some of our interviewees were already a single parent for many years. We think that an assumption can be made that single mothers adapt to their family status as a single parent family. However, this adaption has to be mapped with longitudinal data rather than the cross-sectional differences that can be observed now. Especially due to the socio-economic gradient and intersectionality, we have to handle this with care. Nevertheless, the inclusion of single parents that were single for a longer time enabled us to study single parents that had renegotiated their position in between motherhood ideology and the workplace as well.

A fourth remark was that we could make a more firm statement regarding the observed socio-economic gradient of single mothers’ perspective at the labour market. This is a very valuable comment. We are used to be careful with stating our findings as qualitative researchers but we rephrased our comment and added some sentence to increase the awareness of the indicators for the socio-economic gradient. We also added this in the limitations.

A fifth concern was about the interpretation of the increase of female labour force participation as anticipation, a precedent or a selective antecedent. We are aware that this was not clear in our initial manuscript. Therefore, we rephrased the paragraph, with an improved structure regarding the concepts of anticipation, precedents and selective antecedents.

Finally there were two smaller remarks. A first one was regarding the concept of maternal gatekeeping. Both reviewers have different opinions on the issue of maternal gatekeeping. As reviewer 1 was enthusiast on the inclusion of the concept in order to contextualise the internalised maternal role of parents and reviewer 2 thought maternal gatekeeping was not at its place. We have chosen to keep the topic both in the introduction and in the discussion with more context on the research topic to integrate it better in the rest of the research.

A second concern was about a quote in which an interviewee talked about the importance of Dutch language skills at the workplace. It was not clear why this was important but Dutch was the main language of our interviewees and an important asset on the labour market. We added a footnote here to clarify why this was important.

We reworked the manuscript based on your remarks and hope we improved the article for publication in Social Sciences